# An Overview of Managed Aquifer Recharge in Brazil

**Tatsuo Shubo [1,*], Lucila Fernandes [2] and Suzana Gico Montenegro [2]**

[1]  Departamento de Saneamento e Saúde Ambiental, Escola Nacional de Saúde Pública,
   Fundação Oswaldo Cruz, 21040-361 Rio de Janeiro, Brazil

[2]  Centro de Tecnologia e Geociências, Universidade Federal de Pernambuco, 50670-901 Recife, Brazil;
   lucila.araujo@gmail.com (L.F.); suzanam.ufpe@gmail.com (S.G.M.)

*  Correspondence: tatsuo.shubo@ensp.fiocruz.br; Tel.: +55-21-2598-2469

**Abstract:** In order to face the severe climate conditions in semiarid regions, many managed aquifer recharge (MAR) and rainwater storage systems have been implemented by local communities. Governmental programs have helped to propagate the concept of MAR. Based on a systematic review, popular initiatives, current legislation, and research lines and programs were compiled and analyzed. Although the MAR global inventory points to the prevalence of in-channel modifications among ninety MAR sites, the Barraginhas Project alone has been responsible for the construction of more than 500,000 infiltration ponds up to 2013. In urban areas, aquifer recharge initiatives mostly aim to reduce runoff peak flows. In some cases these initiatives have been stimulated by urban drainage public policies. Compared to countries such as the USA and Australia, Brazil is still at an early stage in MAR initiatives and needs to overcome technical, legal, and socio-cultural challenges to adopt MAR approaches, in order to help in facing water security challenges in a future climate change scenario. This article aims to provide an overview of the state of the art concerning technological, scientific, and legal issues around MAR in Brazil and the respective challenges for the adoption of this approach at a national level.

**Keywords:** water security; urban water management; semiarid; Social Technology; Managed Aquifer Recharge; developing countries

---

## 1. Introduction

### 1.1. Historical Background

Although Brazil has a huge water availability, about 30,342 m$^3$/inhab./year in 2015 [1], it is not evenly distributed across the country, with 80% of the surface water concentrated in the Amazon region [2]. Besides this, Brazil has been struggling with many water crises since the beginning of its settlement by Europeans. A priest called Fernando Cardin recorded the first drought in 1583. Since then, more than 120 droughts have been recorded in the northeastern semiarid region alone. A seven-year drought recorded in the 18th century (1720–1727) that struck the region currently known as the states of Ceará, Rio Grande do Norte, Paraíba, and Pernambuco has been considered the worst one on record. During that particular event, most livestock perished, rivers and springs dried up, and widespread starvation devastated the region [3]. The time period between the years of 1877 and 1879, recorded as the hottest and driest of the 19th century, imposed severe hardships and suffering on the local populations. During that period, approximately five hundred thousand people starved to death, and crops and cattle suffered devastating losses. This scenario triggered massive waves of migration of people moving towards coastal cities, bringing a demographic explosion to areas that did not yet have the appropriate infrastructure in place to support these migrations. Poor living conditions, such as the

lack of proper sanitation systems, have been associated with a smallpox epidemic which contributed to the hardships [4].

From the 1980s up to the present time, Brazil has been experiencing many of its worst droughts on record and struggling with their consequences. As a consequence of yet another seven-year drought period (1979–1985), more than 3.5 million people ended up starving to death, with most of the victims being children who perished from undernourishment. Crops and cattle were lost, forcing desperate farm people to loot local markets seeking food [5]. In 2002, Brazil also faced an energy crisis, known as "the apagão" (The Big Blackout), mostly caused by a series of dry periods. In 2007, the northern part of the state of Minas Gerais suffered a fifteen-month dry spell with virtually no rainfall.

In the Brazilian countryside, especially in the semiarid region, there was a lack of rainfall during the time period of 1981–2019. On the other hand, in the northern Region, high rates of rainfall still occur. Climate models show a trend towards increased frequency and intensity of droughts and length of dry periods in the northeast, as already has occurred in some Brazilian regions [6]. From 2012 to 2017 another major drought affected the semiarid region and 2015 was considered the most critical year of that period. Figure 1a shows the average precipitation for the early dry season in Brazil (April–May), from 1981 to 2010, and the Figure 1b shows the total precipitation in 2015. In the Southern Hemisphere, autumn is the transitional period from the wet to the dry season. As can be seen, in 2015, the total precipitation was far lower than the historical average [7]. Although 2017 has not been the driest year in the northeastern region of Brazil, the rainfall amounts there were far below the historic average, and can be counted as an extension of the 2012 drought. During this period (2012–2017), some of the São Francisco River Hydrographic Region gauging stations recorded zero flow, and the flow released by the reservoirs had to be reduced in order to avoid water supply failures. In 2017, 38 million people were affected by drought and 51% (2.839) of all Brazilian municipalities declared a state of emergency [8]. Brasília, the capital of the nation, and São Paulo, the largest and wealthiest city, endured water rationing during this period.

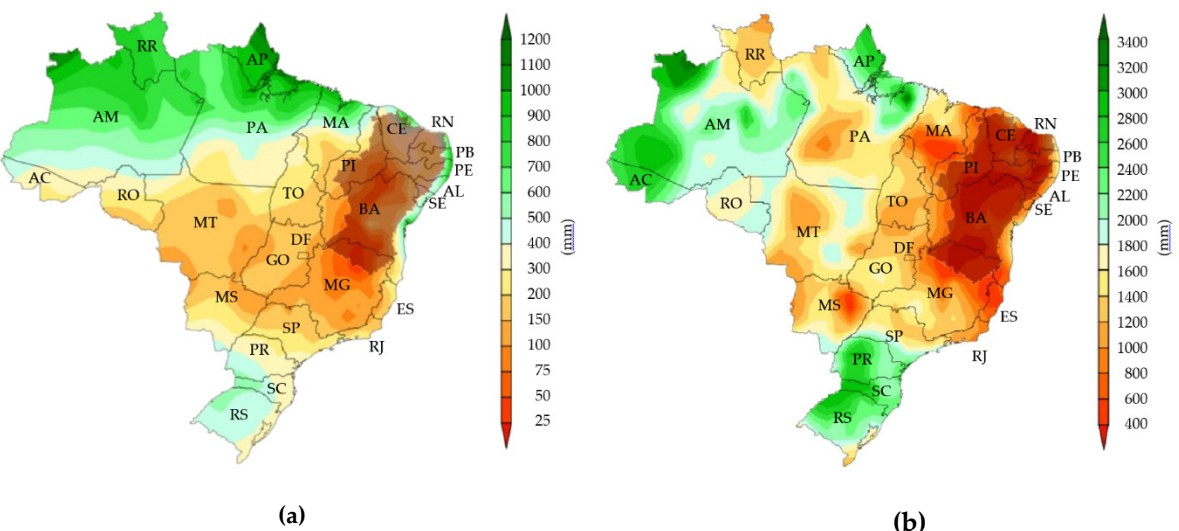

(a)　　　　　　　　　　　　　　　　　　　　　　　　　　　　(b)

| AC: Acre | AL: Alagoas | AP: Amapá | AM: Amazonas | BA: Bahia | CE: Ceará | DF: Distrito Federal |
|---|---|---|---|---|---|---|

ES: Espírito Santo　　GO: Goiás　　MA: Maranhão　　MT: Mato Grosso　　MS: Mato Grosso do Sul　　MG: Minas Gerais
PA: Pará　　PB: Paraíba　　PR: Paraná　　PE: Pernambuco　　PI: Piauí　　RJ: Rio de Janeiro　　RN: Rio Grande do Norte
RS: Rio Grande do Sul　　RO: Rondônia　　RR: Roraima　　SC: Santa Catarina　　SP: São Paulo　　SE: Sergipe　　TO: Tocantins

**Figure 1.** (**a**) Average rainfall in the early dry season (1981–2010); (**b**) total precipitation in 2015. Source: Adapted from Instituto Nacional de Meteorologia (INMET) [7]. The semiarid region (shaded in the Figure 1a,b) encompasses parts of the States of Alagoas, Bahia, Ceará, Minas Gerais, Paraíba, Pernambuco, Piauí, Sergipe, and the whole State of Rio Grande do Norte.

### 1.2. Water Availability in Brazil

According to the ANA [8], close to 90% of all Brazilian rivers rely on a base flow from aquifers that feed these rivers during dry periods, keeping them perennial. The exception occurs in the northeastern region where the ground is composed of a thin layer of soil and fractured rocks known to be crystalline, unable to feed water back to the rivers.

Although surface water dams are the main plan of action against droughts in the northeast, corresponding to 67% of the solutions adopted by the government [8], in actuality, at the national level, 47% of all municipalities have adopted surface water sources, while 39% consume groundwater, and 14% supply their systems with a mix of both [9]. In 2013 there were 225,868 registered tubular wells across the country. It is estimated, however, that this number could be much larger as a consequence of the proliferation of non-registered wells, possibly close to 477,000 wells in 2013 [10].

In 2014, there were around 21 million people living in the Alto Tietê Hydrographic Basin; 97% of the São Paulo Metropolitan Region population. In this basin, the water demand far exceeds the natural water availability, making it a necessity to import from other basins almost half of all water consumed. As a result of a lack of adequate water resources management, during the 2014–2015 water shortage, the government of the São Paulo State was forced to impose water rationing [11].

Traditionally, investments to fulfill water demands in large Brazilian cities are exclusively allocated to the discovery of new water sources, generally without considering important alternatives, such as water reuse [12]. During the 2014–2015 water crisis, the number of private wells for groundwater extraction increased exponentially as a popular response to water access restrictions, ultimately depleting regional underground water sources.

Groundwater sourcing represents, thus, a very important issue in the national and international context of water management both for rural and as well as urban areas. Given this, the introduction of the managed aquifer recharge (MAR) concept is an innovation in integrated water resources management in Brazil. In a climate change scenario and ever-increasing demand, it is considered both as an adaptation measure regarding extreme events, such as droughts, as well as a mitigation strategy for future water crises.

### 1.3. MAR as Solution

MAR is defined as "the purposeful recharge of water to aquifers for subsequent recovery or environmental benefit" [13]. MAR can take many forms, including recharge weirs, infiltration basins, riverbank filtration, recharge releases from dams, and recharge wells. The applications of MAR have been implemented since the 1950s for various purposes, such as to increase groundwater storage, improve quality, restore groundwater levels, prevent saline intrusion, and increase ecological benefits [14]. In many arid or semiarid areas, where groundwater is usually already overexploited or saline, recharge has the potential of storing excessive runoff, including in fractured rocks aquifers [15]. The most common MAR applications are for maximizing natural storage, representing 45% of the case studies in Australia, 62% in South America, and 84% in Africa. The main application of MAR in Europe is for water quality management, where approximately 200 riverbank filtration schemes are used for the production of drinking water [14].

In China there are reports of canals dug in the middle of the 5th century BC close to rivers in regions periodically flooded by storm water. These channels were intended to facilitate the infiltration of surface water into groundwater, changing the quality of groundwater and transforming saline land into fertile soil [16]. The USA today stands out in MAR capacity, second only to India [17]. Arizona has implemented MAR facilities that are able to recharge up to 173 M m$^3$ of the Colorado River, the water of which is considered a renewable resource. In the same state, another project consists of spreading basins through a flood plain producing an annual recharge of up to 37 M m$^3$ [18]. Arizona has a policy for the specific types of MAR that may be implemented in the state. The Underground Storage and Recovery Act, 1986, and the Underground Water Storage, Savings and Replenishment Program,

1994, provide guidelines for allowing state-supported aquifer recharge. This legislation involves three permissions [17].

This paper aims to provide an overview of the current use of MAR, and the potential and challenges of the adoption of MAR in Brazil, by surveying practices adopted to mitigate the effects of droughts, relevant research, and current policy and legal frameworks.

## 2. Methodology

A systematic literature review was carried out with the goal of understanding the challenges and opportunities of MAR adoption to strengthen integrated water resources management (IWRM) to confront climate changes in Brazil. Initially, the main social technologies (ST) to deal with droughts were identified, which were simple, low-cost, and easily applicable technologies. Information was sought about their main uses and relevant constructive features. A free term search was applied aiming to eliminate any misunderstandings about these technologies, as a function of the idiomatic diversity among those who share the knowledge construction base.

Then, further research was carried out, in order to provide a brief overview of stormwater/rainwater infiltration and retention techniques increasingly put to use in Brazilian urban areas. Considering that the sustainable urban drainage systems (SUDS) are solutions used worldwide, we decided that it is not necessary to delve into these technologies with as much detail as in the national rural ST.

Finally, environmental and groundwater legislation at national and state levels were gathered. In possession of these documents, the keywords managed aquifer recharge, artificial recharge, and groundwater were used to find the laws that provide specific guidelines on this subject.

The data were obtained from papers, conference proceedings, academic theses and, mainly, from official reports of state and federal institutions that handle this topic directly, e.g., Embrapa and Articulation for the Semi-arid (ASA).

## 3. Results

### 3.1. Strategies to Deal with Drought in Brazil

When starting a study in any field of knowledge shared by people whose languages are different, it is essential to define clearly basic concepts and terminology. Thus, in this section, the concepts of MAR are discussed, along with the complementary measures of rainwater and stormwater harvesting techniques found in Brazilian rural areas.

As already defined, MAR is a set of measures that aim to artificially increase the recharge of water in an aquifer. In the Brazilian semiarid region, some of the most common techniques for increasing the natural water reserve are underground dams and infiltration ponds. A detailing of these and other techniques and specific examples applied in Brazil are reported below.

The MAR Global Inventory, available on the International Groundwater Resources Assessment Centre (IGRAC) portal, has gathered ninety MAR applications located in Brazil divided in eight specific MAR types: ditch and furrow, dug well–shaft–pit injection, excess irrigation, induced bank filtration, infiltration ponds and basins, rooftop rainwater harvesting, subsurface dam, and trenches [19]. According to this inventory, MAR solutions are mostly concentrated in the northeast region, as can be seen in Figure 2 below. With respect to the specific MAR type, this inventory points to a predominance of subsurface dam technology (64%), mainly located in northeastern Brazil, 100% of which aim to maximize natural storage to be used for agricultural purposes. In reality, what has been called "subsurface dam" technology in Brazil should be named "underground dam", which is divided into two main types: submerged dams (Costa & Mello type) and submersible dams [20,21]. These methods will be described later. The main influent sources used in Brazil are river water, representing 54% of the cases and stormwater, representing 40%. Regarding the final uses of MAR, 60 applications are used for agriculture, 20 cases use water for domestic use, 8 have ecological uses, and only 2 applications are for research purposes [19].

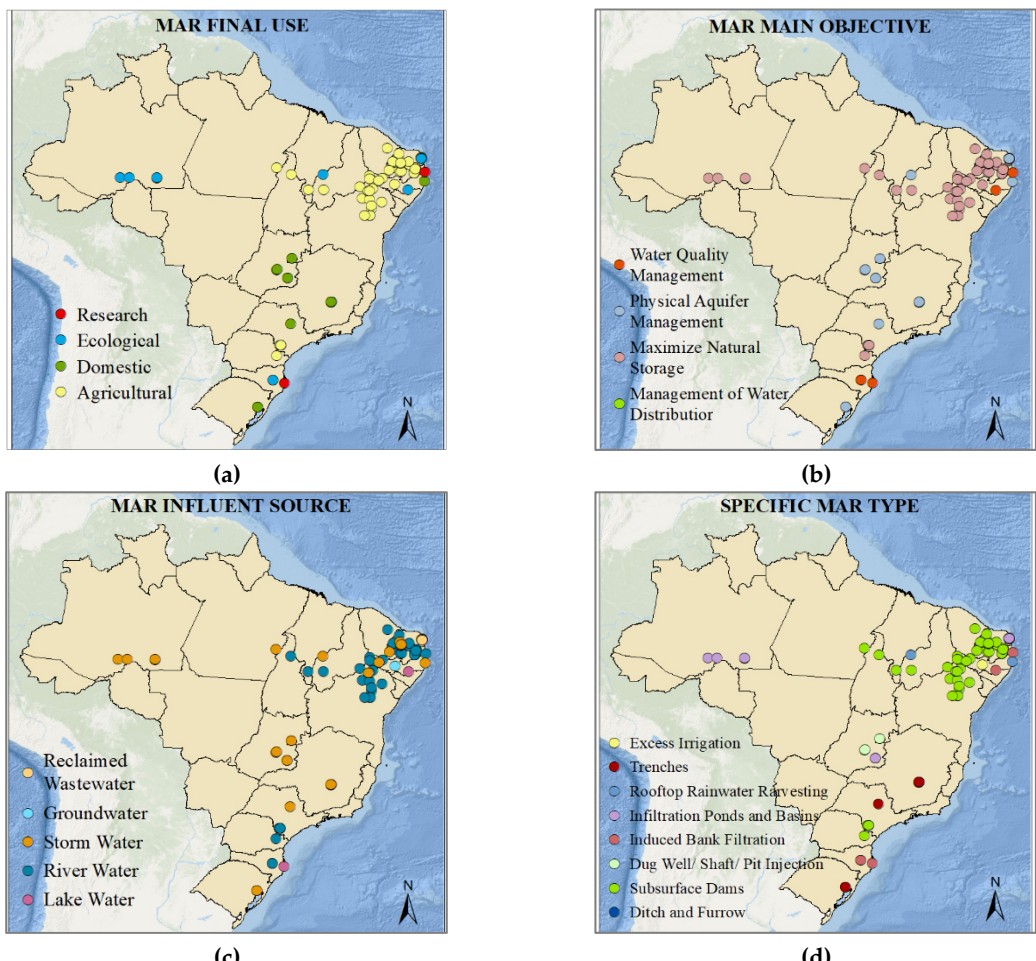

**Figure 2.** Localization of managed aquifer recharge (MAR) solutions in Brazil, sorted by (**a**) MAR final use; (**b**) MAR main objective; (**c**) MAR influent source; (**d**) Specific MAR type. Source: Adapted from International Groundwater Resources Assessment Centre (IGRAC) website [19].

The applications of underground dams in the Brazilian semiarid region have been very important for transforming the reality of the region's farming population, as they are sustainable and easy to apply technologies. In Alexandrina-RN, the construction of an underground dam of approximately 2.0 ha provided a significant increase in the production of maize, beans and rice, allowing an increase in family income through the sale of the surplus [22]. In Paraíba, the Local Development Training Project constructed two underground dams in two communities in the municipality of Texeira. In one of the communities it was possible to harvest fruit even in the dry season. In addition, the owners reported a better quality of the harvested fruits, as bigger and better looking. The participation of the community itself in the construction of these dams is highlighted [23]. Silva et al. [24] analyzed the use of four underground dams, one in Pernambuco, one in Paraíba, and two in Bahia. In all of them, the importance of underground dams was verified for the food security of the families, as well as the food security of their animals.

The effort in water quality monitoring varied from site to site. Samples of water taken from 8 underground dams located in the states of Pernambuco and Bahia were analyzed [25]. Among them, six dams presented low salinity and low sodicity water, which made them suitable for crop irrigation. The other two dams presented water with some risk for use in irrigation, requiring careful monitoring and soil and water management actions.

Infiltration ponds are widespread in Brazil and can be found in thirteen states, and the Federal District as well [26]. Up to 2013, the Barraginhas Project alone has been responsible for implementing

around 500,000 small ponds, all around the country [27]. According to Embrapa Generated Technologies Impacts Evaluation Report, the project has also been responsible for ensuring water and food security and income generation for thousands of families in the semiarid region. Social Technology has also promoted environmental benefits such as soil conservation/restoration, headwaters recovery, and groundwater replenishment [28].

The program P1MC—A Million Cisterns ("Um Milhão de Cisternas") is a program promoted by ASA, whose objective is to promote and ensure access to drinking water for communities in the semiarid region. Based on the principle of stocking up in times of plenty to have enough in times of shortage, this project was awarded in the Future Policy Award in 2017. The project started in 2003; and the goal of building 1 million cisterns was achieved in 2014. Another program of great relevance for living with the semiarid region was the P1+2—One Land and Two Waters Program ("Uma Terra Duas Águas"), whose objective was to increase water security, as well as to promote land management, food security, and income generation [29].

Most of the rural makeshift schemes implemented to deal with drought in Brazil are low-cost measures focused on storing rainwater and stormwater in buried or semi-buried tanks, aiming to ensure small farmers' food and water security. A large portion of these methods are recorded in Government Documents, in free magazines distributed by NGOs and their websites, and referenced in scientific papers.

In the next section, there is a description of some examples of MAR technology that are based on water infiltration into the soil. Following this, some technologies are presented that are considered as MAR, but do not involve aquifer recharge, being technologies for storing water in buried or semi-buried tanks.

### 3.1.1. MAR based Solutions in Brazil

- Infiltration Pond (Small Dam)

Infiltration ponds are well-known in Brazil as small dams (Barraginhas). The main objective in utilizing this ancient technique is soil restoration and conservation. The first experiment with a small dam was performed at Embrapa Milho and Sorgo, in Sete Lagoas—Minas Gerais, 1991. The success of this experiment triggered the widespread use of this technique, extending it to the Brazilian semiarid region with the objective of helping small farming communities to deal with degradation and water shortages [30]. Aragão [31] states that, although widely used in Brazil, there is a lack of studies on the most suitable areas that allow the best performance of this technique.

Small dams are small half-moon-shaped dams. Their dimensions range from 1.5 m to 2.0 m in depth, and 15 m to 20 m in diameter. They are scattered and successively constructed in the main thalwegs of pastureland and degraded fields (Figure 3a), as well as along roadsides, aiming to prevent soil erosion by surface runoff [32]. Figure 3b shows a cross section of a small dam scheme. In steeper and dry thalwegs the dam cross-sectional shapes need to be trapezoidal. On smooth level thalwegs and roadsides, a triangular shape is the format usually utilized [30,33]. The dams are equipped with spillways in both sides to squirt around the excess of runoff, protecting the structure [34].

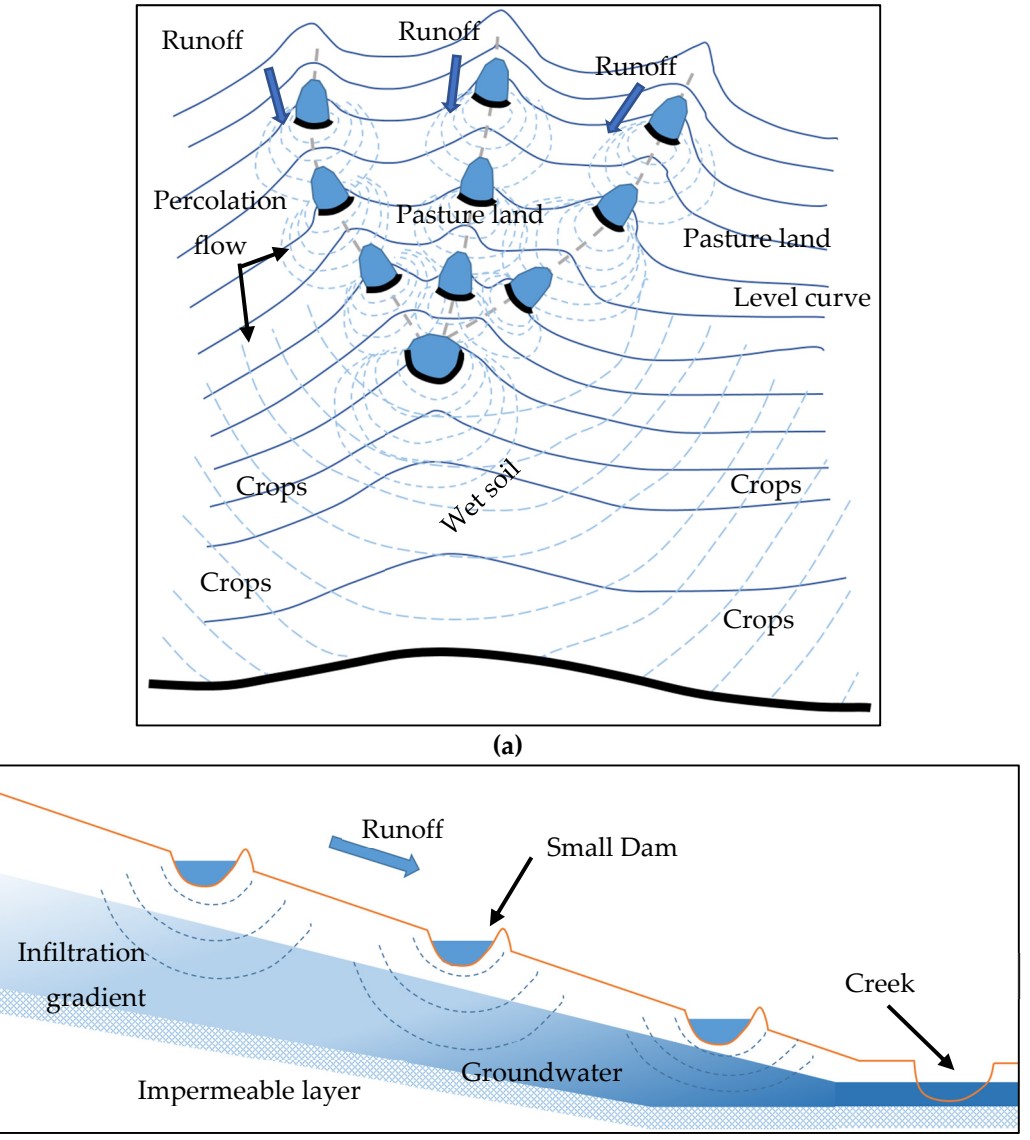

**Figure 3.** Small Dam scheme: (**a**) example of a scattered located small dam scheme; (**b**) cross section of a small dam scheme.

Environmental features, such as topography, soil type, and land use as well, are fundamental elements that affect its location and functioning. A multicriteria analysis carried out on pre-existing small dams, to check if the locations are the most suitable, concluded that slopes less than 3% are not suitable for the implementation of small dams, slopes between 3% and 8% are moderately adequate, and slopes between 8% and 20% are the most suitable [31]. However, field practice points to the fact that slopes greater than 12% should be avoided [35]. Regarding land use and vegetation cover, anthropized areas with sparse vegetation are the most appropriate [31].

Although the soil features such as infiltration and percolation rates are critical for the analysis of rainwater infiltration systems [36], there is a lack of investigation correlating these points to the efficiency of the small dams. A study carried out on eight small dams built in the north of the state of Minas Gerais has highlighted that the recharge capacity increases with soil porosity. Regarding maintenance, silting processes are the main cause of efficiency loss. Around over eleven years of use, the radiuses of the small dams have decreased 1.26 m on average, and the depth loss was up to 1.32 m. Based on these data, the study recommends that every five years, the owners remove the silting from the small dams until recovering the original dimensions (diameter 6 m, depth 1.5–2.0 m) [37].

In order for this technique to be effective, annual precipitation can be up to 1800 mm. Depending on soil type, each small dam can percolate from 800 m$^3$ up to 1200 m$^3$ in a wet season [31]. The notion of successive dams actually helping with retention of pollutants, and the soil acting as a filtering medium, and thus improving water quality, is approached as a consequence of the application of this method, not as its intended goal.

- Underground Dam

An underground dam is an ST that allows rainwater to be stored under riverbeds during the rainy season, making it available for the dry season. Although simple, its construction has to comply with some technical requirements regarding the local where it is built: the alluvium must be predominantly sandy; the slope has to be as level as possible; the depth of the impermeable layer must be greater than 1.5 m; the construction site must be at the narrowest part of the riverbed; and the river head should be avoided, where there is less water. Regarding water quality, low salinity rates are essential to make its implementation feasible [21].

The core idea of its operation is to restrict the flow of the alluvial aquifer by building an impermeable transverse septum, thus raising the level of the upstream water table. In Brazil, there are two types of underground dams suitable to local features: the submerged type, also known as the Costa & Mello type and submersible type. Both of them make use of a buried impermeable septum to restrict the underground flow and are equipped with an Amazon type well to allow the use of the accumulated water in the saturated zone. To build the septum, a trench is dug down to the impervious layer. Then, a plastic blanket is placed over the septum and covered with the excavated material, to block the groundwater flow [21,38,39].

The first type, known as a Costa & Mello-type underground dam (submerged dam), is suitable for the bed river of temporary creeks where the thickness of the sedimentary layer is greater than 1.5 m. This style of construction uses an impermeable septum that is totally buried, retaining only the groundwater flow, making the water table in the alluvium rise upstream of the barrier. There is no physical constraint to the runoff [21,38,39].

There are records of this type of technology in India, Turkey, and Japan, for both irrigation and saline intrusion containment [40–42]. In the Brazilian semiarid region, mainly in the states of Pernambuco, Ceará, and Rio Grande do Norte, this is one of the most applied techniques to deal with water shortages [39]. Figure 4 shows a schematic ground plan (Figure 4a) and cross-section (Figure 4b) of a Costa & Mello type underground dam.

In the second type of underground dam, the submersible underground dam, apart from the buried septum, there is another one made of rocks, bricks, or clay over the riverbed. This barrier makes the superficial flow spread over the land, creating a water pond that lasts up to two to three months after the end of the wet season [21,38,39]. The process of lake formation generates a gradual accumulation of sediments, increasing the thickness of the soil upstream of the dam, thus providing an increase of the storage capacity over time, as happens in sand dams [20,43]. However, some authors warn that it only happens in some cases [44]. This technique is suitable for small rivers and water pathways. This dam over the riverbed is equipped with a spillway made of concrete to spill over excess water and preserve the barrage above the ground, limiting the water level. Upstream, close to the dam, an Amazon-type well is built to recover water for irrigation and for other uses, such as livestock water supply when the water level falls below to the ground level [21,38,39]. Figure 5a shows a schematic ground plan of the submersible underground dam. Figure 5b shows a schematic cross-section of the submersible underground dam.

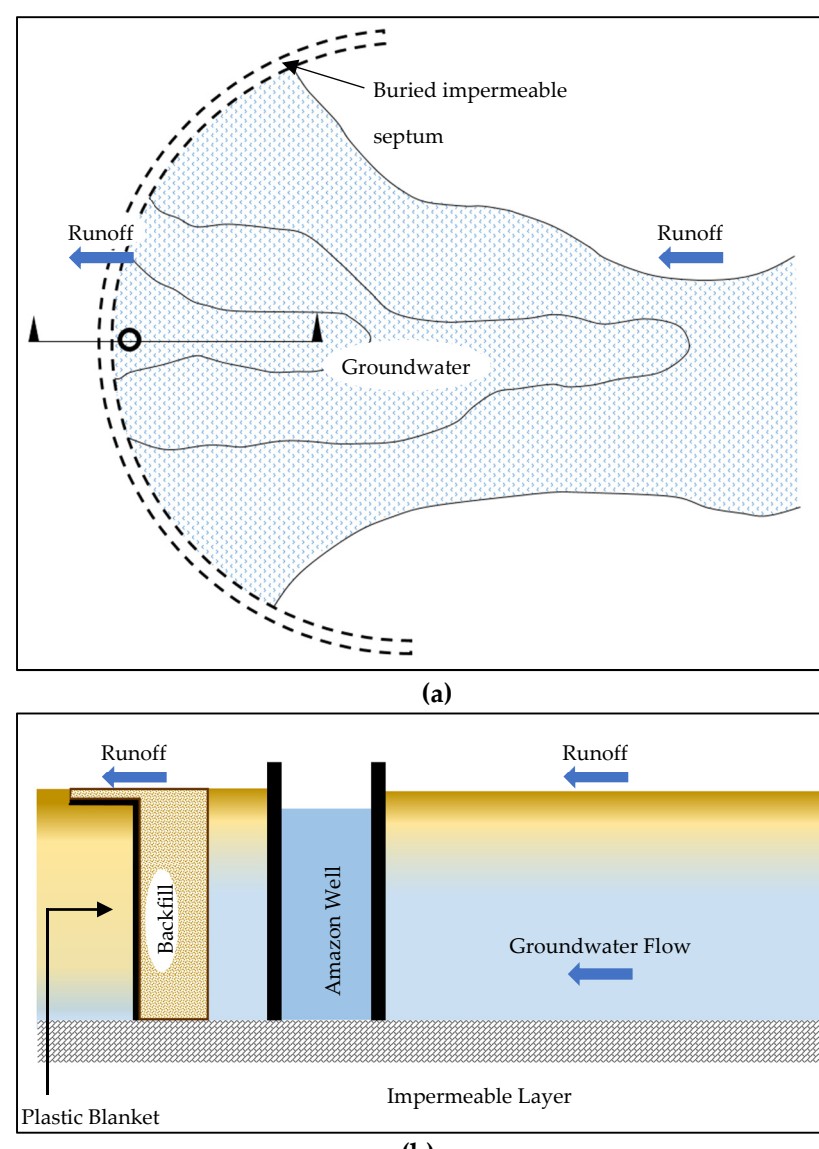

**Figure 4.** Costa & Mello type underground dam—submerged dam (**a**) Ground plan; (**b**) Cross section.

To implement an underground dam, a set of several factors affects the costs. Among them, the length of the impermeable septum, the raw material, the depth of the impermeable layer, and the workforce available. Based on an underground dam with a septum length of 100 m, using a plastic blanket, and a maximum depth of up to 3.5 m, a study performed by Semiárid Embrapa (CPATSA)/Farming Brazilian Research Company (EMBRAPA) estimated the costs to be around 1300 USD if using heavy machinery [45].

- Dry Well (Caixa Seca)

This is a rainwater harvesting method used to reduce soil erosion, preventing unpaved road deterioration and the silting of rivers and streams. These structures are normally built in a series connected by trenches dug along the roadsides. Based on the builder's empirical knowledge of runoff speed, they are associated with ditches dug diagonally to the axis of the roads, aiming at reducing the surface runoff speed and to covey the rainwater to the dry wells [46]. For safety reasons in the reduction of erosion risks, Bertoni and Neto [47] have recommended that the spacing between the dry wells should be within specific limits, as shown in the Table 1 below.

**Table 1.** Maximum spacing between the Dry Wells as a function of road slope.

| Road Slope (%) | Distance between Dry Wells (m) |
| --- | --- |
| 5–10 | 50 |
| 10–15 | 30 |
| >15 | 20 |

Source: adapted from Bertoni and Lombardi Neto [47].

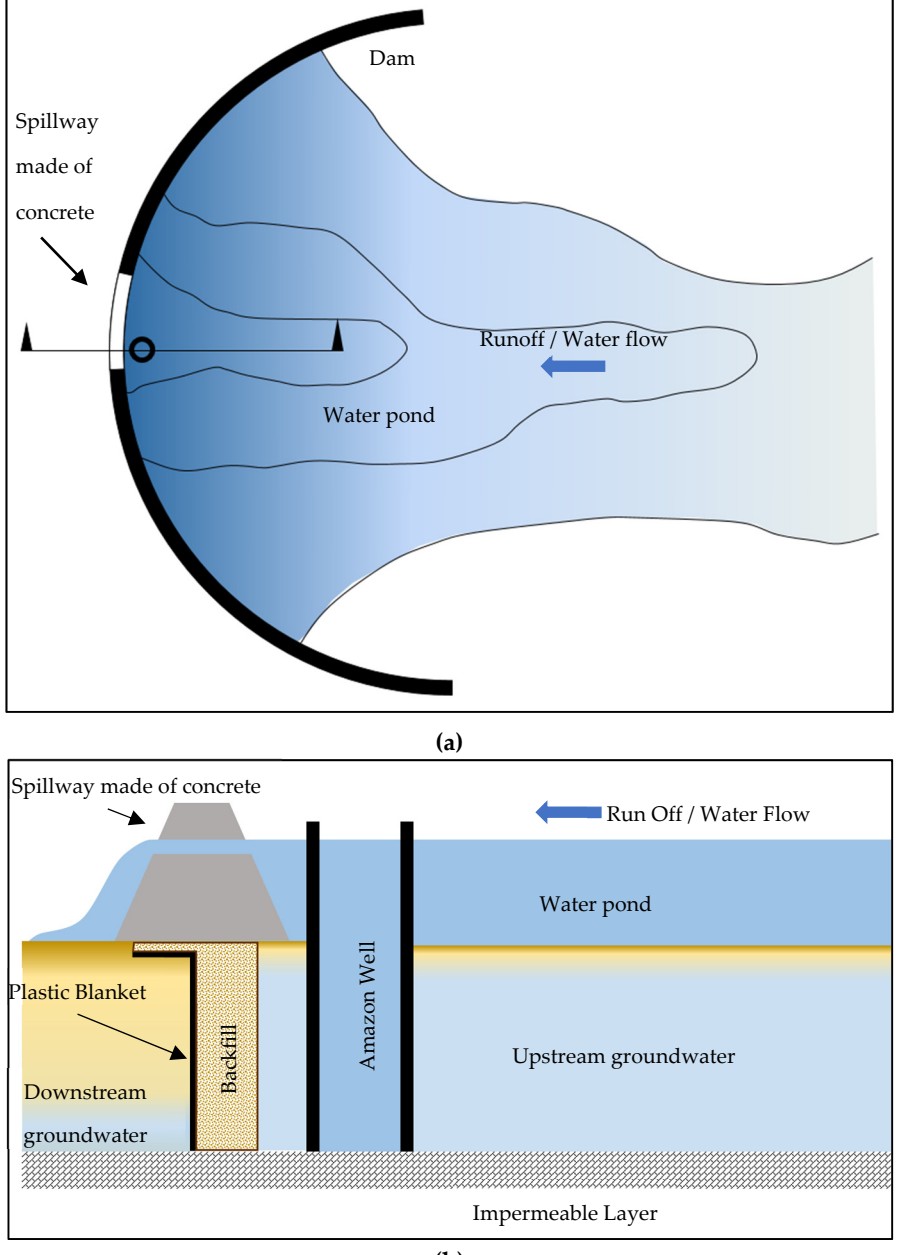

**(a)**

**(b)**

**Figure 5.** Submersible underground dam (**a**) ground plan; (**b**) cross section.

The dry well dimensions are defined in the field, based on the builders' experience, never being less than 1 m deep, 1 m wide, and 1 m long (1 m$^3$). Caixa Seca translates literally to English as dry box. The trenches are 0.30 m in depth, and their width is defined by the width of a hoe. These structures also function as infiltration devices, helping to distribute the water into the ground [48]. Whenever

the silt in a dry well reaches 50% of its volume, the silt must be removed [49]. Figure 6 shows a dry well scheme.

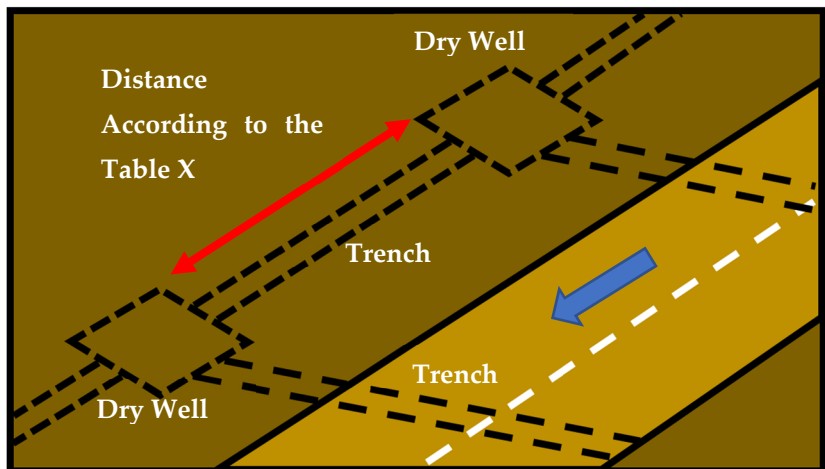

**Figure 6.** Dry well (caixa seca) scheme.

- Bank Filtration

Bank filtration is a simplified water treatment technique developed in Europe more than a hundred years ago [50]. It consists of groundwater abstraction by a constructed well located close to a river or a lake, aiming to induce a groundwater gradient, forcing the infiltration of the surface water towards the well, thereby improving water quality [51]. Depending on the soil, underground, and water source features, bank filtration could be the only water treatment before a final chlorination step, or at least used as a pre-treatment [52]. The use of this technique is not limited to the Brazilian semiarid region. It can also be found in the Southeastern Region, especially in the State of Santa Catarina, in the south of Brazil. The bank filtration scheme is shown in Figure 7.

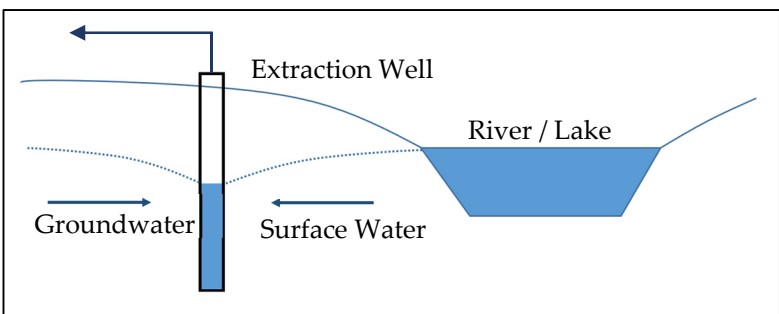

**Figure 7.** Bank filtration scheme.

### 3.1.2. Other Solutions

- Rainwater Tank

A rainwater tank is a covered, semi-buried cistern connected to the house gutters and equipped with a first flush device. Its capacity ranges between 16,000 and 21,000 L. The most common storage capacity is the 16,000 L with a 3.45 m diameter and 2.4 m depth. Since it is covered and made from concrete, it is impervious, preventing evaporation and debris contamination [53,54].

- Boardwalk Cistern (Cisterna calçadão)

This method was conceived of as a way to help scattered small farming communities not being served by a large-scale hydraulic infrastructure as a way to help them sustain their food production in their backyards.

It is a covered, semi-buried reservoir 6.4 m in diameter and 1.8 m in depth. Connected to a 200 $m^2$ concrete floor by a 100 mm pipe, this system is able to collect and store around 52 $m^3$ of rainwater even under yearly rainfall rates around 350 mm. Since it is covered and made of concrete, it is impervious, preventing evaporation, and debris contamination.

The harvesting area known as boardwalk (calçadão) is surrounded by a small concrete fence, and has a slight slope towards the cistern. The linking device between the boardwalk and the cistern is provided with a small sedimentation tank to avoid particles entering the cistern [55].

- Stormwater Tank

This tank has a 6.2 m diameter, a depth of 1.8 m, and is a covered, totally buried, concrete-plate made cistern, capable of storing up to 52 $m^3$ of stormwater. The main difference between this technique and the boardwalk cistern is the catchment area. In this model, there is no specific design for the harvesting area. It must be implemented in a slightly sloped ground (<5%) where a water stream naturally flows. A vegetated catchment should be used to avoid the erosion of coarse sediments. This ST uses two serially-placed sedimentation tanks before the water reaches the cistern, in order to remove sand and small stones from the harvested water [56].

- Retention Trench

The standard retention trench (barreiro trench) is a narrow, deep, inverse-trapezoidal shaped, dug reservoir, capable of storing around 500 $m^3$ of stormwater. The trapezium's larger base measures approximately 24 m with the smaller base at around 16 m, the width at 5 m, and the depth varying from 3 m to 5 m. It should be implemented in the runoff natural pathway, where the terrain slope is as low as possible, avoiding silting and consequent storage volume losses. Its feasibility must be checked by at least three survey boreholes along to the reservoir axis with the objective of identifying the impermeable layer depth [57].

*3.2. Early MAR Initiatives in Brazil: Projects and Researches*

In larger Brazilian cities, the main rainwater and stormwater management concern is the runoff peak flow attenuation and flooding avoidance. Towards this, various sustainable drainage techniques have been encouraged, but only a few of these focus on stormwater infiltration. Among them, the most widely used techniques are rain gardens, infiltration trenches, and permeable pavements [58]. None of these have water quality control concerns before infiltration and nor water recovery methods.

Nevertheless, a few MAR initiatives have been evaluated, mainly in the academic environment, as a way of improving integrated water resource management processes. They contribute to the assessment of the technical feasibility and were not implemented on a large scale. Okpala [59] carried out the first specific MAR academic assessment in Brazil in 2010. The study assessed the soil aquifer treatment (SAT) feasibility at the Governador André Franco Montoro International Airport (State of São Paulo), in order to help the Brazilian National Airport Infrastructure Company—Infraero to discover an alternative water source for the growing water demand. In this pilot experiment, the most suitable area was chosen from a group of previously assessed areas. The undisturbed samples taken at the unsaturated layer of the selected area were characterized and subsequently assembled into special testing columns, through which secondary treated effluent was infiltrated. Another set of undisturbed samples were taken from a second area similar to the first one. Based on the results, the authors concluded that for SAT be feasible, the first layer of the soil should be removed or replaced by coarse sand for adequate treatment.

Rayis [60] has assessed water quality requirements and costs for MAR implementation in the São Paulo Metropolitan Region as a response to the 2015 water crises. The most suitable treatment process, recharge methods, and water-treated sewage quality features were based on international experiences, such as in the cities of Shafdan (Israel), Atlantis (South Africa), Sabadell (Spain), and Adelaide (Australia). The study adopted influent water quality standards based on the legislation from the US and Spain that specify the required sewage treatment plant effluent treatment level, which is the tertiary treatment level for nitrogen removal. The MAR unitary operation cost was estimated at 1.41 USD/m$^3$, almost twice the cost of a water mains unitary operation.

The BRAMAR Project, cooperation between Brazil and Germany, intended to help face water shortage by planning and preparing MAR operation schemes. Thus, the Gramame river coastal basin, in the state of Paraíba, was chosen as a study area, aiming to assess the soil infiltration capability and the necessary treatment efficiency of the effluent from the stabilization pond. For this purpose, a 10 mm h$^{-1}$ average input flow was infiltrated into two undisturbed soil columns collected from the study area. Over 72 days, the researchers assessed a set of six physicochemical parameters (BOD$_5$, COD, DOC, TSS, NH$_3$, and NO$_3$). The results showed a reduction in organic matter, suspended soil, and ammoniacal nitrogen greater than 60%. Clogging problems were observed, and the feed procedures were changed. From the 42nd day, wet and dry cycles were implemented aiming to restore soil infiltration capability [61].

A master thesis developed at Campina Grande Federal University aimed to propose a transition from an existing non-managed aquifer recharge reality to an intended MAR scenario in the semiarid region [62]. The study area was the Surucucu alluvial aquifer, located at Paraíba River Basin, in the Sumé Municipality, Paraíba State. The lithologic characterization of the study area was carried out based on data of 117 intrusive investigation boreholes made by the BRAMAR Project. The water table level was monitored through a set of 40 wells distributed along 12 km of the Surucucu riverside, from April 2016 to October 2017. From May 2016 to October 2017, the study has used the chloride ion as a tracer to indicate aquifer contamination by sewage. The results showed that the chloride concentrations kept in high levels, especially in the urban area due to the lack of sanitation. Although decreasing along with the underground flow, the results showed the negative impacts of the non-managed aquifer recharge, pointing out to the risk of salinization. The study has proposed a set of actions to work from non-managed aquifer recharge towards MAR, such as to infiltrate treated sewage by using infiltration ponds, where there is no restriction of space, and to recharge surface water resources when it is possible by using an aquifer storage transfer and recovery systems (ASTR).

### 3.3. Groundwater Legal Framework

This section aims to present the main groundwater legal documents from both federal and state levels. Although not the same, artificial recharge is the closest in meaning to the MAR expression. Therefore, it was used as a keyword in the place of MAR when searching on the internet for documents related to the Brazilian legal framework for MAR. From now on, the acronym "MAR" will be used in place of "artificial recharge".

### 3.3.1. Federal Level

From a set of seventeen groundwater legal documents at the federal level, four (23.5 %) mention MAR in their content. Up to the early 2000s, there was no legislation clearly addressing MAR. In 2001, the Water Resources National Council Resolution n° 15 encouraged municipalities to adopt MAR. In the following year, Water Resources National Council Resolution n° 22 established that withdraw and recharge estimates should be included in the water resources plans. In 2008, the Environment National Council Resolution n° 396 permitted establishment of MAR to avoid saline intrusion, providing that there were no changes in water quality, and established water quality mandatory monitoring. At the end of the same year, the Water Resources National Council Resolution n° 92 made prior authorization and mandatory monitoring a condition of aquifer recharge (Table 2).

**Table 2.** Legal framework at the federal level.

| Legal Documents Addressing Underground Water | 17 |
|---|---|
| **Underground Water Legislation Addressing MAR** | **4** |
| **Legal Document** | **Article** |
| CONAMA N° 396/2008 | Art. 23 Allows MAR to avoid saline intrusion Art. 25 Made water quality monitoring mandatory |
| Resolution n° 15, 2001/01/11 | Art. 6 Encourages MAR |
| Resolution n° 22, 2002/05/24 | Art. 3 Establishes discharge and recharge estimate mandatory in Water Resources Plans |
| Resolution n° 92, 2008/11/05 | Art. 8 Conditions aquifer recharge on prior authorizationArt. 10 VIII Makes water quality monitoring mandatory |

### 3.3.2. State Level

Of all state water resources laws that address underground water, around a fifth (20.3%) mention MAR. Most of them have been promulgated by the states located in the semiarid region, and follows the federal level regulations. Concerning treated wastewater, the Tocantins State legal framework prohibits its discharge into groundwater, although it allows MAR under technical, economic and sanitary assessment, and prior authorization by the Tocantins' Nature Institute. The States of Pernambuco, Ceará, and Maranhão define MAR clearly as water injection through underground dams or injection wells. Santa Catarina State's definition of MAR is generic, defining it as any intentional infiltration technique. It should be highlighted that, among several states, only Pernambuco and Ceará encourage MAR adoption by citizens and companies through rebate schemes on sanitation taxes. Both Pernambuco and Ceará States condition water withdraws to natural recharge features maintenance or MAR. Table 3 shows the basic principles of the state laws addressing MAR.

**Table 3.** Legal Framework at the State Level.

| Legal Documents Addressing Underground Water | | 74 |
|---|---|---|
| **Underground Water Legislation Addressing MAR** | | **15** |
| **State** | **Legal Documents** | **Basic Contents Related to MAR** |
| Pará | State Law n° 6.381, 2001/07/25 | The State Water Resources Council allows MAR under technical, economic, and sanitary assessment, preserving groundwater quality |
| Tocantins | State Law n° 1.307, 2002/03/22 | The State Water Resources Council allows MAR under technical, economic, and sanitary assessment, preserving groundwater quality |
| | Decree n° 2.432, 2005/06/06 | Prohibits treated wastewater discharge into groundwater, although |
| | | The State Water Resources Council allows MAR under technical, economic and sanitary assessment, and prior authorization by Tocantins' Nature Institute |
| Roraima | State Law n° 547, 2006/06/23 | The State Water Resources Entity along the with Watershed Council allow MAR under technical, economic, and sanitary assessment |
| Pernambuco | Decree n° 20.423, 1998/03/26 | Defines MAR clearly as water injection through underground dams or injection wells |
| | | Encourages MAR adoption by citizens and companies through rebate schemes on sanitation taxes |
| | | Conditions water withdraws to natural recharge features maintenance or MAR, and prior authorization |
| Ceará | State Law n° 14.844, 2010/12/28 | Supports or carries out MAR projects to ensure groundwater quality and quantity |
| | | Defines MAR clearly as water injection through underground dams or injection wells |
| | | Encourages MAR adoption by citizens and companies through rebate schemes on sanitation taxes |
| | Decree n° 31.077, 2012/12/12 | Conditions water withdraws to natural recharge features maintenance or MAR, and prior authorization |
| Maranhão | Decree n° 28.008, 2012/01/30 | Defines MAR clearly as water injection through underground dams or injection wells |
| | | Conditions water withdraws to natural recharge features maintenance or MAR, and prior authorization |
| Bahia | State Law n° 11.612, 2009/10/08 | Supports or carries out MAR projects to ensure groundwater quality and quantity |
| Alagoas | State Law n° 7.094, 2009/09/02 | Encourages MAR projects to ensure groundwater quality and quantity |
| | | Conditions water withdraws to natural recharge features maintenance or MAR, and prior authorization |
| Piauí | State Law n° 5.165, 2000/08/17 | The State Water Resources Council allows MAR under technical, economic, and sanitary assessment, preserving groundwater quality |
| Minas Gerais | State Law n° 13.771, 2000/12/11 | The State Water Resources Council allows MAR under technical, economic, and sanitary assessment |
| Espírito Santo | State Law n° 6.295, 2000/07/27 | The State Water Resources Council allows MAR under technical, economic, and sanitary assessment |
| Santa Catarina | Resolution CERH n° 02, 2014/08/14 | Defines MAR as any intentional infiltration technique |
| | | The State Water Resources Council allows MAR under technical, economic, and sanitary assessment, preserving groundwater quality |
| Rio Grande do Sul | Decree n° 42.047, 2002/12/26 | Conditions MAR to prior authorization by State Agencies |

## 4. Discussion

Brazilian folk wisdom has been the generator of several initiatives to the fight against droughts. At least two of them have been developed and applied only in Brazil: small dams (Barraginhas) and dry wells (Caixa Seca). However, regarding water quality and volume monitoring, there are no systematic records that allow a scientific MAR approach. Although the MAR global inventory points to the prevalence of in-channel modifications (underground dams) over the infiltration ponds and basins, there is no registration of small dams (Barraginhas) on the platform, resulting in underreporting of this type of technology. However, there are governmental reports that state there have been more than 500,000 small dams constructed. It is possible to estimate the number of schemes implemented and costs by mining grey literature, but the reliability of these information should be checked in field research. Some of these technologies have been implemented into government programs, such as P1MC and P1+2. It must be highlighted that some STs, such as rainwater tanks, stormwater cisterns, boardwalk cisterns, and retention trenches may be misinterpreted as an aquifer recharge practice. Although using buried or semi-buried reservoirs, these STs use the impervious tanks to store, not to infiltrate the harvested rainwater. In a large portion of the northeastern region, the soil features, such as salinity and low infiltration rates, have led to these technologies instead of aquifer recharge [62]. Despite being widely encouraged by legal frameworks at both federal and state levels, and being used all around the country, MAR technologies have no water quality monitoring programs in Brazil due to both the lack of a custom of monitoring and government underfunding.

Although federal and state legislation cites water reuse and MAR, there is no clear link between them [60]. In addition, the use of alternative sources of water is generally viewed with some suspicion, as a result of an intricate set of social and institutional barriers resulting from the perception of risks related to the various possible uses of this water [60]. This lack of information and long-term studies is one of the factors that hinder the development of appropriate legal framework supporting on MAR in Brazil. In view of this, projects such as the international cooperation Brazil-Germany (BRAMAR), which helped to face water shortage by planning and preparing MAR operation schemes, should be encouraged, and their results should be further disseminated to promote the acceptance and encouragement of this type of technology.

The main concern related to rainwater/stormwater urban management is the runoff peak flow attenuation and flood avoidance by sustainable drainage techniques. Most of these techniques focus on rainwater retention close to where it precipitates. None of them have aquifer recharge for further water recovery purposes. Consequently, they cannot be considered MAR applications. This delay in adopting MAR initiatives, as happens also in India, induces the risk of loss of economic and social benefits [63]. However, since 2010, there have been some academic MAR assessment initiatives for both urban and rural areas as well. Regarding economic assessment of MAR in urban areas, preliminary assessment points to the MAR unitary operation cost to be almost twice the unitary water mains operation cost (US$ 1.41/m$^3$ versus US$ 0.75/m$^3$) [60]. It should be highlighted that "[t]he averaged costs of water supplies from four desalination plants in eastern and southern Australia built since the 'millennium drought' [64] to secure capital city supplies is more than 10 times the long-run marginal costs of normal supplies in those cities" [65].

## 5. Conclusions

The semiarid region of Brazil and big cities have been identified as the main areas that need strategies to combat water scarcity and measures that ensure water security. Identifying aquifers that are suitable for MAR application sand the availability of water sources for recharge are strategic actions to enable the selection of projects that present best cost benefits and that are alternatives for the use of traditional sources.

Although there are some studies being developed in the academic environment, Brazil is still at an early stage in MAR initiatives and needs to overcome technical, legal, and socio-cultural challenges to adopt MAR. Adoption of MAR should be considered a strategy for facing future droughts under

a climate change scenario [65]. However, the lack of awareness concerning MAR solutions and non-specific local policies linking MAR schemes to water demand have delayed its development in Brazil. In this sense, it is necessary to identify areas, such as the semiarid region in the northeast, urban settlements, and water-intensive agricultural lands, where the demand for water overcomes the availability, threatening water security. In these areas, the government should encourage the search for aquifers suitable for MAR, on which its adoption could be a more cost-effective alternative than traditional sources, with the perspective of improving the integrated management of water resources.

Most Brazilian MAR scheme records are dispersed in grey literature, such as non-governmental organization (NGO) websites and governmental reports. The MAR Global Inventory is, therefore, a helpful initiative to gather them in a reliable source. However, the majority of the information must be checked in the field by research projects.

As has happened in a number of countries where MAR schemes have been widely adopted, there are no consistent regulations for the implementation of MAR in Brazil. Seeking to establish a MAR regulation framework, to be based on the Australian guidelines, is a sound and less difficult attempt to provide principles for safe implementation of MAR schemes [66]. Taking advantage of state autonomy in groundwater management [60], guidelines on MAR suitable for local features must be developed to help improving integrated water resources management in Brazil based on scientific approaches.

**Author Contributions:** T.S.: Writing—original draft; L.F.: Writing—review & editing; S.G.M.: Writing—review & editing. All authors have read and agreed to the published version of the manuscript.

**Funding:** This research was partially funded by CNPq (Conselho Nacional de Desenvolvimento Científico e Tecnológico) through the DIGIRES project, grant no. 400128/2019-5. Also, the authors would like to acknowledge the financial support from CNPq for the PQ research grant, and from FACEPE (Fundação de Amparo à Ciência e Tecnologia do Estado de Pernambuco) for granting PhD scholarship.

**Acknowledgments:** The authors would like to thank the Vice-Presidency of the Environment, Attention and Health Promotion - Fiocruz Brasil, for supporting the research and Rogério Silva, for his contribution to the English review. Our special thanks to Warish Ahmed for kindly transfer his waiver quote to publish this manuscript. Finally, we thank the reviewers and editors for their valuable comments and suggestions.

**Conflicts of Interest:** The authors declare no conflict of interest.

## Abbreviations

The following abbreviations are used in this manuscript:

| | |
|---|---|
| ANA | National Water Agency |
| ASA | Articulation for the Semi-arid |
| ASTR | Aquifer Storage Transfer and Recovery |
| BRAMAR | Brazil Managed Aquifer Recharge |
| CPATSA | Semiárid Embrapa |
| Embrapa | Farming Brazilian Research Company |
| Infraero | Brazilian Airport Infrastructure Company |
| IGRAC | International Groundwater Resources Assessment Centre |
| IWRM | Integrated Water Resources Management |
| MAR | Managed Aquifer Recharge |
| NGO | Non-governmental Organization |
| P1+2 | One Land, Two Waters Program |
| P1MC | One Million Cisterns Program |
| SAT | Soil Aquifer Treatment |
| SPMR | São Paulo Metropolitan Region |
| ST | Social Technologies |
| SUDS | Sustainable Urban Drainage Systems |

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
