# Peer review of "An Overview of Managed Aquifer Recharge in Brazil"

_water, doi:10.3390/w12041072_

Round 1

Reviewer 1 Report

Managing aquifer recharge (MAR) is a very innovative approach to be discussed in Brazil. A country, as the manuscript points out, is world-wide known by its water resources richness, but still suffers by water shortage in several places and periods along the last decades.

The manuscript is well readable, the arguments regarding MAR solutions for increasing water supply are relevant, but the main scientific question is not clearly addressed along the text and some specific aspects need to be confronted before the publication of this manuscript. Considering the comments listed below and the scientific potential of this study, I highly recommend a major revision of it.

Specific comments

The background literature is clearly articulated and provides useful information regarding drought and water shortage precedents in Brazil, but considering the proposal of the study (MAR Perspectives in Brazil) it still needs to be equilibrated with MAR concept, aspects and some examples around the world.

The MAR example developed in Tucson, Arizona, US is extremely interesting and applicable to the context of the manuscript. Because it is a kind of arid region, with MAR as a water supply solution. See: https://www.tucsonaz.gov/water/recharged-water

The Portuguese terms must be translated in some way, e.g. Barraginhas = Small dams, then it will be better understood by Water Journal readers and been able to be propagated.

It needs to better cite and specify the references along the MAR solutions. For example, what is the reference of “Caatinga” and “Costa e Melo” types of underground dam? In addition, the EMBRAPA references should be written properly, as proceedings, book chapters, reports, etc, not just as websites.

The MAR solutions should be followed by schemes (Figures) representing the engineering background of each solution. Then, the reader will be able to better differ the solution types among each other.

I think the lines 225 and 235 are missing citations of Okpala nd Rayis.

Some arguments of Topic 3.2 can be carried out to Introduction as a State of Art of MAR initiatives in Brazil.

Since the manuscript objective is: “This paper aims to provide an overview of the potentialities and challenges in the adoption of MAR in Brazil, by surveying practices adopted to mitigate the effects of droughts, relevant researches, and current policy and legal frameworks.”, I suggest a topic 3.3 connecting the arguments of Topic 3.1 (MAR Solutions) and 3.2 (Legal framework) that clearly conducts the reader to “potentialities and challenges in the adoption of MAR in Brazil”.

Some empirical information related to the increasing of water supply associated with aquifer recharge could be strongly helpful to sustain the manuscript proposal.

The abbreviation MAR in the title is not appropriated and helpful for paper searchers. I suggest something like: “An overview of managed aquifer recharge in Brazil”

Author Response

Dear Reviewer,

First of all, we would like to thank you for your remarks. All of them were very useful to make this manuscript more robust and readable. We thoroughly worked on each item to comply with the high level of these comments. The changes are highlighted on the World file with the "Track Changes" function. As requested, we are providing a point-by-point detailing of the changes.

We submitted the manuscript to check the English correctness by a native speaker.

We are available to answer any questions.

Best regards

Point 1: The background literature is clearly articulated and provides useful information regarding drought and water shortage precedents in Brazil, but considering the proposal of the study (MAR Perspectives in Brazil) it still needs to be equilibrated with MAR concept, aspects and some examples around the world.

 Response 1: We have divided the introduction into three parts: 1.1 Historical background; 1.2. Water availability in Brazil; 1.3 MAR based solutions. In the third one, we described the MAR concept, giving an overview of MAR all around the world, and in Brazil as well.

Point 2: The MAR example developed in Tucson, Arizona, US is extremely interesting and applicable to the context of the manuscript. Because it is a kind of arid region, with MAR as a water supply solution. See: https://www.tucsonaz.gov/water/recharged-water.

Response 2:  Thank you for this tip. It was very useful. We have inserted some brief information about this example of MAR in the sub-topic 1.3.

Point 3: The Portuguese terms must be translated in some way, e.g. Barraginhas = Small dams, then it will be better understood by Water Journal readers and been able to be propagated.

Response 3: We have translated the terms, keeping the name in Portuguese between brackets to allow the readers finding references in Portuguese literature.

Point 4: It needs to better cite and specify the references along the MAR solutions. For example, what is the reference of “Caatinga” and “Costa e Melo” types of underground dam? In addition, the EMBRAPA references should be written properly, as proceedings, book chapters, reports, etc, not just as websites.

Response 4: We rewrote the entire item, inserting more appropriate references, as suggested.

Point 5: The MAR solutions should be followed by schemes (Figures) representing the engineering background of each solution. Then, the reader will be able to better differ the solution types among each other.

Response 5: Thank you for this observation. We have drawn schematic drafts and inserted in the text to allow a better understanding.

Point 6: I think the lines 225 and 235 are missing citations of Okpala nd Rayis.

Response 6: We have corrected this point by inserting the proper references.

Point 7: Some arguments of Topic 3.2 can be carried out to Introduction as a State of Art of MAR initiatives in Brazil.

Response 7: We maintained the information in Topic 3.2 but added some information about the State of Art of MAR in Brazil in the introduction to enrich the topic.

Point 8: Since the manuscript objective is: “This paper aims to provide an overview of the potentialities and challenges in the adoption of MAR in Brazil, by surveying practices adopted to mitigate the effects of droughts, relevant researches, and current policy and legal frameworks.”, I suggest a topic 3.3 connecting the arguments of Topic 3.1 (MAR Solutions) and 3.2 (Legal framework) that clearly conducts the reader to “potentialities and challenges in the adoption of MAR in Brazil”.

Response 8: Thank you again for the comment. We have accepted the proposal. We did it in the Discussion Section.

Point 9: Some empirical information related to the increasing of water supply associated with aquifer recharge could be strongly helpful to sustain the manuscript proposal.

Response 9: We have addressed this issue in the section 3.1. Strategies to Deal With Drought in Brazil. The entire item related to the underground dam (in the 3.1.1 – MAR Solutions) has been rewritten, and a new paragraph was inserted to include information on costs, unit water storage capacity, and crop yield as well.

Point 10: The abbreviation MAR in the title is not appropriated and helpful for paper searchers. I suggest something like: “An overview of managed aquifer recharge in Brazil”.

Response 10: We changed the title as suggested.

Reviewer 2 Report

The paper aims to organise the practice of managed aquifer recharge in Brazil. The authors conducted a comprehensive literature and policy review to indicate scientific and regulatory requirements to be implemented to face the environmental change and the increasing water consumption.

A number of techniques and their practical use are discussed. Some of them require technical and operational fine-tuning in order to improve their efficiency. A few methods including soil aquifer treatment are very promising due to a need for a conjunctive water storage and treatment. A vast majority of techniques require water quality monitoring to be in place. Increasing public perception, international collaboration and more focused funding are likely to address these needs.   

Specific comments:

There are a few types of underground dams used in Brazil. It would be good to provide some basic technical information: size, stored volumes, etc. 

Line 150 - improper reference

Lines 179-180 - -se names of states rather than regions to comply to fig. 1

Lines 225 - it is not clear what Okpala means. Please provide a correct reference in the first sentence.

Line 253 - lack of referencing.

Line - 261 - not a proper reference

Line 277 Chart or Table?

I strongly suggest editing by a native English speaker.

Author Response

Dear Reviewer,

First of all, we would like to thank you for your remarks. All of them were very useful to make this manuscript more robust and readable. We thoroughly worked on each item to comply with the high level of these comments. The changes are highlighted on the World file with the "Track Changes" function. As requested, we are providing a point-by-point detailing of the changes.

We submitted the manuscript to check the English correctness by a native speaker.

We are available to answer any questions.

Best regards

Point 1: There are a few types of underground dams used in Brazil. It would be good to provide some basic technical information: size, stored volumes, etc.

 Response 1: Thank you for this comment. The entire item related to the underground dam (in the 3.1.1 – MAR Solutions) has been rewritten, and a new paragraph was inserted to include information on costs, unit water storage capacity, and crop yield as well.

Point 2: Line 150 - improper reference.

Response 2: We rewrote the item related to the underground dam, this point has been corrected.

Point 3: Lines 179-180 - -se names of states rather than regions to comply to fig. 1.

Response 3: Thank you for your comment. We have drawn a map of Brazilian Geographic Regions, highlighting the Semiarid Region and the parts of the States that compose it.

Point 4: Lines 225 - it is not clear what Okpala means. Please provide a correct reference in the first sentence.

Response 4: The reference has been inserted.

Point 5: Line 253 - lack of referencing.

Response 5: The reference has been inserted.

Point 6: Line - 261 - not a proper reference.

Response 6: The reference has been corrected.

Point 7: Line 277 Chart or Table?

Response 7: Thank you for your observation. It is a Table. The item has been corrected.

Reviewer 3 Report

It is a generalist article about MAR in Brazil generalities, which fits very well with this special edition´s targets and ISMAR 10 philosophy.

The article is averaged for the journal and the development is up to expectations.

Please, check the detailed comments exposed next:

Notes:

7- Despite its worldwide water resource wealth reputation > i do not have this perception of Brazil and have never had, even doubt this statement is shared by most of the technicians willing to read this article. I´d soften this thought ?

7-19. Have authors considered a brief comparison with other countries´ state-of-the-art. It would be an asset either by some indicators or simply based on the author´s vision.

23- The intro includes a large background. May a sub-chapter be inserted?

24-37. Any histogram or figure to make more visual this large amount of literature? I really miss to get a general idea in a simple bird´s sight with the distance between periods of drought´s occurrence. Regarding future models and forecasts, I have my doubts to include the results, but they would be really visual. There is room for a figure in this position, the article is not long.

Top-right corners: page 2 of 4 when there are 12 pages. Future edition will take care of these mistakes.

In my humble opinion the figure fits the text, but is not the best unless being accompanied by the one suggested about the evolution and expected trends.

Please specify the hydrological character of the year 2015: wet, dry, medium? Is it representative?

Check English correction, pls.

69-74. I miss reference [7]

88 could be > is (reviewer´s personal opinion for your consideration)

88 adaptation: are you excluding mitigation?

This reviewer misses a paragraph including previous references of MAR in Brazil, not for the S-O-T-A regarding climate and trends but specifically for MAR. The MAR catalog of IGRAC… Probably they might be grouped in rural areas, SUDS or classified according to the type of MAR or design.

Later I realize authors are grouping the experiences in the results paragraph, anyway the previous references are missed.

I think this is important and marks a difference with the papers already accepted in this special edition, breaking the line.

ST? It is clear to me but perhaps not to everyone

93-103. with the due respect, it is an index or declaration of intentions rather than a methodology description. Please, enlarge the text fitting the content to the title.

When authors mention it is the cheapest for hard rocks areas, are also considering the excavations? Only for the surface regolith? Is this really cheaper than constructions for detritic areas? Any range might be specified with specific figures?

165: “In the steeper road sections” any data or figure might be provided? What is steeper? from 10%, 15%...?

does it fit a Haefeli bucket design or is it made according to common sense?

Figure 3: what are we seeing? (apart from where?)

I miss a map with of Brazil with the distribution of the exposed Tech Sols. The article is rather scarce in figures, maps, graphics and resources in general. Is it a specific study on SUDS? Where are these places? Not everybody knows…

235-242. The economic part might be separated from the title: 3.2. Rainwater/Stormwater Management and Aquifer Recharge Initiatives in Brazil

243-252. Isn´t this something to be included in the no existing MAR SOTA in Brazil paragraph? Is it something to be done (objective) or results from BRAMAR project?

253-261. Same comment than for the previous excerpt. It speaks about monitoring for a specific site related to an specific project. Are there any monitoring guidelines at Brazilian level?

May be you could extract and mention two or three specially remarkable and useful for the community. I think it is not 3.2.1 but rather 3.3 or even 4, as the topic completely changes in respect to 3.2. Please, reconsider the order you are exposing different and unrelated topics, specially in the 3.2. paragraph. Are regulations results from this paper?

263-266. It happens in the whole south American countries. MAR and AR are separated at a very expert level, but for S.A. governments (ANAs, CONAMAs): Artificial recharge = MAR, and that is a reality that we must accept although we desire something different.

330-332. Please, insert one comma, at least.

It is like a list of desires. Please, separate in different lines or points and include comments about the degree of advance in each line of action with personal comments based on the already exposed about the importance of each. Even a ranking of the most urgent actions to be done would enrich the article very much.

As general comments for both, authors and editors:

This reviewer misses a paragraph including previous references of MAR in Brazil, not for the S-O-T-A regarding climate and trends but specifically for MAR. The MAR catalog of IGRAC…

The article is scarce in figures, maps, graphics and resources in general. It is about Brazil and the unique map is for the distribution of precipitation. Please, consider enrich the article with usual resources to gain expressivity and visibility…

In summary, the article fits well the special edition of the journal, the topic is interesting and very welcome, but some more work from authors is required to be in the average level, specially on maps, graphics and general resources apart from simple photographs and maps captured from references.

It does not provide great advances at scientific level, but it covers a gap in the S-O-T-A and the platform to be included is perfect, but the quality of the paper must improve according to the comments already recommended.

The general distribution of the information should be reconsidered, especially for the results section.

In my opinion the papers must be included in “accept after minor changes”, but the changes are not that minor. I encourage authors to seek sometime to improve the general quality of the paper heeding some of the reviewer´s recommendations, for theirs and everyone´s sake! At this moment it is still far from final acceptance. At least two pages more would be required to satisfy all the detected gaps, what is reasonable for the final extension of the paper. Out of 12 pages, almost three are references!!!

Author Response

Dear Reviewer,

First of all, we would like to thank you for your remarks. All of them were very useful to make this manuscript more robust and readable. We thoroughly worked on each item to comply with the high level of these comments. The changes are highlighted on the World file with the "Track Changes" function. As requested, we are providing a point-by-point detailing of the changes.

We submitted the manuscript to check the English correctness by a native speaker.

We are available to answer any questions.

Best regards

Point 1: 7- Despite its worldwide water resource wealth reputation > i do not have this perception of Brazil and have never had, even doubt this statement is shared by most of the technicians willing to read this article. I´d soften this thought?

Response 1: We have softened this statement as suggested.

Point 2: 7-19. Have authors considered a brief comparison with other countries´ state-of-the-art. It would be an asset either by some indicators or simply based on the author´s vision.

Response 2: We insert an item (MAR as solution) on the introduction, on which we quickly addressed the MAR concept and applications. In this paragraph, we also addressed the Tucson example and cited the share of MAR applications in Brazil.

Point 3: 23- The intro includes a large background. May a sub-chapter be inserted?

Response 3: As suggested, we divided the Introduction in three sub-chapters:  Water shortages: historical background; Water availability in Brazil; MAR as solution.

Point 4: 4-37. Any histogram or figure to make more visual this large amount of literature? I really miss to get a general idea in a simple bird´s sight with the distance between periods of drought´s occurrence. Regarding future models and forecasts, I have my doubts to include the results, but they would be really visual. There is room for a figure in this position, the article is not long.

Response 4: Thank you for your comment. We have checked this point and made some changes to clarify what we have wanted to show to the readers. In this sense, we would like to enlighten the statement about the trend of "increasing the frequency and intensity of droughts and length of dry periods in the Northeast" is not of our authorship. We are citing the reference nº 7. Thus, to clarify this point, we have gathered precipitation data since 1981 and created a figure to show this trend (figure 1). This figure is complemented by figures 1 (a) and (b). Figure 1(a) shows the average precipitation in the early dry season. Figure 1(b) shows the total precipitation in 2015. As can be seen, low rainfall rates reached the Southeast Region. This scenario is not a common one. We have made some changes to the text as well (lines 64-67 & 69-71). We have also prepared a map with the Brazilian Geographic Regions and the location of the semiarid region. If you consider it is necessary, we can insert it.

Point 5: Top-right corners: page 2 of 4 when there are 12 pages. Future edition will take care of these mistakes.

Response 5: Thank you for this observation. We have checked and corrected this issue by using the Word’s tool.

Point 6: In my humble opinion the figure fits the text, but is not the best unless being accompanied by the one suggested about the evolution and expected trends.

Please specify the hydrological character of the year 2015: wet, dry, medium? Is it representative?

Response 6: We have considered this point related to Point 4. Thus, we jointly checked these two points. Please, check if this point has been corrected above.

Point 7: 69-74. I miss reference [7]

Response 7: We have checked all the references by the end of the review and corrected this point as well.

Point 8: 88 could be > is (reviewer´s personal opinion for your consideration)

Response 8: Thank you for your opinion. We accepted your suggestion, stating that MAR is an adaptation action to face climate changes.

Point 9: 88 adaptation: are you excluding mitigation?

Response 9: Thank you for your comment. Maybe we did not make us understand when writing the sentence. We understand that adaptation aims to create mechanisms to lessen vulnerability, and mitigation acts on the root of the problem. In this sense, we have rewritten the statement explaining that MAR is an adaptation measure regarding climate change and a mitigation measure regarding the water crisis.

Point 10: This reviewer misses a paragraph including previous references of MAR in Brazil, not for the S-O-T-A regarding climate and trends but specifically for MAR. The MAR catalog of IGRAC… Probably they might be grouped in rural areas, SUDS or classified according to the type of MAR or design.

Later I realize authors are grouping the experiences in the results paragraph, anyway the previous references are missed.

I think this is important and marks a difference with the papers already accepted in this special edition, breaking the line.

Response 10: Thank you for your comment. We have checked item and we agree its missing an introductory paragraph. Thus, when we divided the Introduction, we have inserted a sub-chapter “1.3 – MAR as solution”, on which we have included some references of MAR in Brazil and in the World as well.

Point 11: ST? It is clear to me but perhaps not to everyone

Response 11: Thank you for your observation. At the end of the manuscript, previously to the References, there is a list of abbreviations on which we give the meaning of all abbreviations. In the methodology chapter, we described what social technology is.  

93-103. with the due respect, it is an index or declaration of intentions rather than a methodology description. Please, enlarge the text fitting the content to the title.

Response 11: Thank you for your comment. We have checked and rewrote the entire item fitting it to the title, as requested.

Point 12: When authors mention it is the cheapest for hard rocks areas, are also considering the excavations? Only for the surface regolith? Is this really cheaper than constructions for detritic areas? Any range might be specified with specific figures?

Response 12: Thank you for this remark. We have rewritten the item related to underground dams, reorganizing the concepts and introducing explanatory figures. Regarding the costs, we inserted a paragraph on which we presented the average cost instead of stating it is cheaper or not.

Point 13: 165: “In the steeper road sections” any data or figure might be provided? What is steeper? from 10%, 15%...?

 Response 13: Thanks for your questioning. We have reviewed this section. We have sought to better describe the empirical features of the construction process. We also inserted a table with the slopes of the road and the corresponding distances between the dry boxes. We provided a schematic draw of the dry well as well.

Point 14: does it fit a Haefeli bucket design or is it made according to common sense?

Figure 3: what are we seeing? (apart from where?)

 Response 14: Thank you for your comment. We revised the text clarifying this point. We also provided a schematic draw of dry boxes location.

Point 15: I miss a map with of Brazil with the distribution of the exposed Tech Sols. The article is rather scarce in figures, maps, graphics and resources in general. Is it a specific study on SUDS? Where are these places? Not everybody knows…

Response 15: Thank you for your appointment. We have rewritten the Item 3.1 - Strategies to deal with drought in Brazil. We also inserted a figure to explore MAR by classifying on final use, main objective, influent source, and MAR type. We hope it works for a better understanding of this point.

Point 16: 235-242. The economic part might be separated from the title: 3.2. Rainwater/Stormwater Management and Aquifer Recharge Initiatives in Brazil

Response 16: Thanks for the comment. Unfortunately, we do not have enough data about the economic aspects to support the discussion in a specific chapter. Moreover, this specific paragraph gathers the main results of a Master Thesis in which the water quality requirements and the economic aspects have been assessed. We changed the title of the Item 3.2 (Rainwater/Stormwater Management and Aquifer Recharge Initiatives in Brazil -> MAR initiatives: projects and researches) to better fit with the content.

Point 17: 243-252. Isn´t this something to be included in the no existing MAR SOTA in Brazil paragraph? Is it something to be done (objective) or results from BRAMAR project?

Response 17: Thank you for this appointment. This paragraph presents the results of one of the projects of the BRAMAR. We have considered that the results of the thesis and pilot-scale projects are the state of the art of MAR in Brazil. Thus, we inserted the BRAMAR initiatives in the results of our manuscript.

Point 18: 253-261. Same comment than for the previous excerpt. It speaks about monitoring for a specific site related to an specific project. Are there any monitoring guidelines at Brazilian level?

Response 18: Thank you for your comment. There is no monitoring guideline at the Brazilian level. The study has used the chloride ion as a tracer to indicate sewage aquifer contamination. We have inserted more specific information to clarify the aim of the study.

Point 19: May be you could extract and mention two or three specially remarkable and useful for the community. I think it is not 3.2.1 but rather 3.3 or even 4, as the topic completely changes in respect to 3.2. Please, reconsider the order you are exposing different and unrelated topics, specially in the 3.2. paragraph. Are regulations results from this paper?

263-266. It happens in the whole south American countries. MAR and AR are separated at a very expert level, but for S.A. governments (ANAs, CONAMAs): Artificial recharge = MAR, and that is a reality that we must accept although we desire something different.

Response 19: Thanks for this contribution. You are absolutely right. We have separated these topics as suggested. We inserted the Item 3.3 Groundwater Legal Framework.

Point 20: 330-332. Please, insert one comma, at least.

Response 20: Thanks for this appointment. We have rewritten the conclusion to make it more readable.

Point 21: It is like a list of desires. Please, separate in different lines or points and include comments about the degree of advance in each line of action with personal comments based on the already exposed about the importance of each. Even a ranking of the most urgent actions to be done would enrich the article very much.

Response 21: Thanks for this appointment. We have rewritten the conclusion to make it more readable, and we inserted some possible actions to enhance the conclusion.
